# The Relationship between Emotional Intelligence, Obesity and Eating Disorder in Children and Adolescents: A Systematic Mapping Review

**DOI:** 10.3390/ijerph18042054

**Published:** 2021-02-20

**Authors:** Emanuele Maria Giusti, Chiara Manna, Anna Scolari, José M. Mestre, Tamara Prevendar, Gianluca Castelnuovo, Giada Pietrabissa

**Affiliations:** 1Psychology Research Laboratory, Istituto Auxologico Italiano IRCCS, 20145 Milan, Italy; e.giusti@auxologico.it (E.M.G.); gianluca.castelnuovo@unicatt.it (G.C.); giada.pietrabissa@unicatt.it (G.P.); 2Department of Psychology, Catholic University of Milan, 20123 Milan, Italy; chiara.manna02@icatt.it; 3Faculty of Psychology, San Raffaele University, 20132 Milan, Italy; anna.scolli@gmail.com; 4INDESS (Institute of Developmental and Social Sustainability), Department of Psychology, University of Cadiz, 11519 Cádiz, Spain; 5Department of Psychotherapy Science, Sigmund Freud University, 1000 Wien, Austria; tamara.prevendar@gmail.com

**Keywords:** emotional intelligence, eating disorders, body image, obesity, children, adolescents, youth, mapping review, clinical psychology

## Abstract

Eating and weight disorders often develop early in life and cause a long-standing significant health burden. Given the documented role of emotional intelligence (EI) in shaping the body image and predicting the onset of eating disorders, knowledge of the mechanisms involved in EI among youth is fundamental to designing specific interventions for screening and prevention of obesity and eating disorders (EDs). The present systematic mapping review was aimed to explore and quantify the nature and distribution of existing research investigating the impact of EI on EDs in young people. A systematic search for relevant articles was conducted using PubMed, Scopus, PsycINFO and Web of Science databases. The Appraisal tool for Cross-Sectional Studies (AXIS) was used to assess the included studies’ methodological quality. The included studies’ results were mapped based on stratification by age groups (children, preadolescents, and adolescents), population (clinical vs. non-clinical) and disordered eating outcomes. Nine studies were included, supporting the association between EI and body image dissatisfaction, ED risk and bulimic symptomatology, but not with anorexic symptoms. Research on children and clinical populations was scant. Further studies are needed to deepen the role of EI in the genesis and maintenance of EDs.

## 1. Introduction

Childhood obesity and accompanied disordered eating attitudes and behaviours are some of the most significance health challenges of the twenty-first century [1,2,3]. Research has consistently revealed that childhood obesity and disordered eating attitudes are each associated with numerous physical health issues that track into adulthood, as well as the increased risk of developing eating disorders (EDs), including anorexia nervosa (AN), bulimia nervosa (BN) and binge eating disorder (BED) [4], during adolescence [5].

These outcomes have been shown to share sociocultural, psychological (perception of ideal body weight, personality factors and individual temperament) and behavioural risk factors in children and adolescents [6]. Notably, a negative body image would have a pivotal role in predicting eating and weight problems in adolescents, owing to the critical age-related period they are facing [7,8]. Several studies have found an association between emotional states and body satisfaction in adolescent girls [9,10]. The appropriate knowledge and regulation of one’s own emotion are a protective factor against the onset of a negative body image [11,12,13]. In this context, research agrees that the concept of emotional intelligence (EI) is comprehensively able to capture the complex set of dimensions featuring the individuals’ way of processing emotional information and behaving in highly emotional situations [14]. The literature describes two predominant conceptualisations of EI: ability EI (AEI) and trait EI (TEI) [15]. On the one hand, AEI refers to one’s cognitive ability to perceive, use, understand and regulate emotion in oneself and others. It should be measured via maximum-performance tests [16]. On the other hand, TEI relates to people’s self-perceptions of their emotional abilities [17], and self-report questionnaires commonly measure it. Correlations between TEI and AEI are low across studies, thus supporting their distinct nature [18]. However, both approaches have complementary explanations of emotional-intelligent behaviour [19]. From AEI literature, individuals with high AEI would be better equipped to deal with emotional difficulties. People with a lower score on AEI are more likely to engage in maladaptive emotional regulation strategies [20], such as disordered eating behaviour. Accordingly, females with an ED have been found to report more maladaptive emotion-focused strategies than their non-clinical counterparts, whilst the ability to regulate emotion has been negatively associated with disordered eating patterns [21,22]. Still, a growing number of studies confirm the association between TEI, body image and disordered eating symptoms in the adult population [23,24,25,26,27], and emerging evidence also shows that an increase in TEI corresponds to decreased healthcare costs, particularly at low education levels [28]. However, there is still little agreement on how emotional regulations relate to eating and weight problems in children and adolescents.

Given the well-known impact of psychological and emotional distress in the misuse of food during the lifetime [29] as well as the documented role of EI in shaping the body image and predicting the onset of ED symptoms, collating the breadth of research activity in this area is fundamental to understanding better the relevant mechanisms involved in emotion dysregulation in children and adolescents, and thus to design specific intervention plans for screening and prevention of obesity and eating disorders.

For such reasons, this paper is aimed to provide an evidence map of the relationship between EI and eating and weight disorders among clinical and non-clinical populations of children and adolescents.

Therefore, a systematic mapping review of the literature was performed to provide an accessible and practical overview of available knowledge on the topic so as to help clinicians incorporate evidence into daily clinical practice and allow the identification of existing research gaps and opportunities for future investigations. The decision to employ this research design is driven by the fact that the available literature on the relationship between EI and eating and weight problems is sparse in terms of methodology and types of posited association between constructs.

## 2. Materials and Methods

### 2.1. Search Strategy

Searches were conducted in PubMed, Scopus, PsycINFO and Web of Science databases from May to June 2020. The search strategies combined key terms and Medical Search Headings (MESH) terms based on the population, predictor, outcome (PPO) framework: (1) population: children, preadolescents, and adolescents; (2) predictor: AEI and TEI; and (3) outcome: obesity, body mass index (BMI) and EDs (including AN, BN, BED, eating disorder not otherwise specified (EDNOS) or other specified feeding or eating disorder (OSFED)) and body image. Boolean and truncation operators were used to systematically combine more searched terms and list documents containing variations on search terms, respectively [30]. The search syntax was modified, as appropriate, for each database.

### 2.2. Inclusion and Exclusion Criteria

Only original articles that (1) employed a cohort or cross-sectional study design; (2) considered emotion regulation as a predictor variable; and (3) examined the association of EI with obesity, body image and disordered eating outcomes (4) in a sample of participants with a mean age ≤18 years (children, preadolescents, and adolescents) were included. Records were excluded if they considered only biomedical factors as outcome variables. No restrictions were set for the language or date of publication.

### 2.3. Study Selection

Following the search and exclusion of duplicates, two reviewers (authors C.M. and A.S.) independently screened the eligibility of selected records according to the inclusion and exclusion criteria based on the title and the abstract using Rayyan software (Qatar Computing Research Institute, HBKU, Doha, Qatar) [31]. The screened records were then exported to Colandr software (National Center for Ecological Analysis and Synthesis, Santa Barbara, U.S.A) for full-text screening and data extraction [32]. Disagreements were solved by a third reviewer (authors G.P. or E.M.G.).

### 2.4. Assessment of Risk of Bias

The Appraisal tool for Cross-Sectional Studies (AXIS) [33] was used to assess the included studies’ methodological quality. AXIS includes 20 items answered with a yes, no or do not know that address study design appropriateness, reporting quality and risk of bias in cross-sectional studies. In detail, item #1 and item #2 appraise the appropriateness of the study design, items #3–7 reflect the presence of selection bias, item #8 and item #9 are representatives of the presence of measurement bias, items #10–16 reflect the presence of reporting bias, and items #17–20 identify the presence of potential confounding impairing the interpretation of the study findings (see Appendix A).

### 2.5. Data Extraction and Synthesis

Authors C.M. and A.S. independently extracted the following data from included studies using a pre-designed data collection form: (1) first author and year of publication; (2) country; (3) study design (cross-sectional or cohort study); (4) study aims; (5) EI operationalisation (trait EI or ability EI); (6) EI measure; (7) population (clinical or non-clinical); (8) sample size, gender (%) and age (mean, SD); (9) BMI (mean, SD, range); (10) primary outcomes (disordered eating, emotional eating, body image satisfaction, bulimic symptomatology, anorexia nervosa, obesity, tendency to develop an ED, body image perception); (11) measure of primary outcomes; and (12) main results.

We addressed a short synthesis of the main findings of the studies included in the map. The extracted data were then employed to map and provide a description of the available evidence regarding the relationship between EI and eating and weight problems.

## 3. Results

Searches of electronic databases yielded 4303 records, of which 33 were duplicates and 4223 records were excluded based on information from the title and the abstract. The remaining 47 articles were evaluated for inclusion by reviewing their full text and resulted in the exclusion of 38 records for the following reasons: (1) they did not employ cross-sectional/cohort studies (*n* = 5), (2) EI was not a predictor (*n* = 11) or (3) participants had a mean age of ≥18 years (*n* = 22). Finally, 9 articles [34,35,36,37,38,39,40,41,42] were included in the systematic mapping review (Figure 1).

### 3.1. Characteristics of Selected Articles

A description of the included studies is reported in Table 1. The included articles were published between 2007 [35] and 2020 [34,38] in China [39,40], Spain [34,42], Germany [38], France [36], Mexico [41], Taiwan [37] and the United States [35]. All studies employed a cross-sectional design.

The sample size ranged from 79 [36] to 2509 [40] across studies. Except for two records [35,36] that included a sample of females only, and of one study [37] in which females constituted 24.4% of the sample, the samples of the remaining records were generally gender homogeneous.

The mean age ranged from 9.58 [38] to 18.66 [35] years across samples. Five out of nine records did not report the BMI of the sample [34,36,37,40,41], and only one study employed the BMI-for-age percentile as an indicator to measure the size and growth patterns of children and teenagers [38]. Among the remaining articles, the mean BMI ranged from 19.1 kg/m^2^ [42] to 23.2 kg/m^2^ [35].

Notably, three studies [37,39,40] focused on AEI but employed the use of self-report measures instead of maximum-performance tests [16]. Given the absence of reliable findings on the association between AEI and eating and weight disorders, the results of this study only concern the role of TEI in predicting the onset and maintenance of eating pathologies.

### 3.2. Methodological Quality of Included Studies

Three [34,38,41] out of nine studies showed a low risk of bias across the AXIS domains (Table 2). The study design’s appropriateness to the study aims was judged inadequate in three studies [35,39,40] due to the use of a cross-sectional design to assess mediation or moderation models, which has been said to be problematic [43]. Selection bias was detected in three studies due to the high non-response rate [42], inclusion of a sample not representative of the population of interest [35], small sample size and failure to report reasons for the sample size [36]. No measurement or reporting bias was detected in any article. Potential confounding impairing the interpretation of study results was noted in one study [37]. Responses to the individual AXIS questions are reported in Appendix A.

### 3.3. Emotional Intelligence and Eating Disorders

A map describing the results of the included studies stratified by age group and population is reported in Figure 2. This map clearly shows that research on children and preadolescents and research on clinical groups regarding the relationship between EI and weight and eating problems are underrepresented.

No study assessed the association between TEI and weight and eating problems in children. Three studies focused on preadolescents [34,38,42]. In this age group, it was found that TEI was inversely associated with body image dissatisfaction, drive for thinness and bulimic symptoms.

In contrast, research on adolescents was more represented. Like in the other age groups, TEI was found to be negatively correlated with body image dissatisfaction [40,42]. In all these studies, a higher TEI was associated with less eating disorder risk. TEI was also a moderator of the relationship between body dissatisfaction and eating disorder risk [40]. Finally, TEI was associated with bulimic symptoms [35,42] but not with the drive for thinness after controlling for body dissatisfaction [42].

Only one study compared the TEI scores of a clinical population and healthy controls. This study did not find any differences in TEI between adolescents diagnosed with anorexia and healthy controls after controlling for anxiety and depression. It is to be noted that no study assessed the relationship between TEI and obesity or BED symptoms in any age group.

## 4. Discussion

This study aimed to map the research evidence on the relationship between TEI and eating and weight disorders in children and adolescents. The results highlight areas that have received research attention, while simultaneously exposing numerous opportunities for research on eating disorder interventions. Gaps in the current evidence are discussed here with a focus on specific research opportunities.

A total of 9 cross-sectional studies [34,35,36,37,38,39,40,41,42] were selected from the initial 4030 publications obtained from four databases (PubMed, Scopus, PsycINFO and Web of Science). The number of cross-sectional studies published per year on the topic has increased over time, especially in the last three years.

This huge number of studies reveals a growing interest among clinicians and researchers in the role of TEI in predicting obesity and EDs, which may be related to the current need for alternative-integrative actions of care able to achieve a meaningful degree of prevention and protection from these syndromes.

In terms of methodological quality, included studies were rated as at low or high risk of bias. Publications with a low risk of bias were prevalent, as an indicator of the reliability of their findings.

The results of this systematic mapping review showed TEI to be inversely associated with body image dissatisfaction, drive for thinness and bulimic symptoms in both preadolescents and adolescents. Findings are in line with other studies supporting TEI and EDs’ association in youths [12,44,45] and adults [46,47,48,49,50]. Accordingly, a previous study conducted on a sample of college students found an independent association between TEI and body image dissatisfaction [25]. Body image is a multidimensional construct that involves perceptions, behaviours, cognitions and emotions related to an individual’s body and, therefore, represents one of the most robust risk factors for the development of EDs [51,52].

In samples of adolescents, associations were also found between TEI and *ED risks*, suggesting that adolescents with difficulties in TEI are more likely to develop ED symptoms. This was similar to the relationship between TEI and bulimic symptomatology *in* adolescents and preadolescents but to a lesser extent.

Findings are consistent with those of Foye et al. (2019) [46], who demonstrated the role of TEI in the genesis and maintenance of EDs and in straightening of patients’ engagement in treatment. Still, Gardner et al. (2014) [24] argued that both TEI and AEI have a role in different bulimic symptoms, including binge eating, compensatory behaviours, and weight and shape concerns, and AEI is more frequently related to compensatory behaviours.

Although impaired emotional functioning and alexithymia traits are core elements of AN [53,54], no associations were found between TEI and anorexic symptoms in adolescents’ clinical and non-clinical populations [36]. Research confirms that compared to healthy controls, individuals with AN present with an internal-dysfunctional way to carry out accurate reasoning about emotions and use emotions and emotional knowledge to enhance thought recognition and regulate their emotional states [36,55,56,57,58]. This result is consistent with the fact that AN is primarily associated with an inflexible and rigid cognitive style [59,60] that, especially when facing negative moods [61], might trigger perfectionism in the sense of obsessive behaviour to avoid possible adverse effects due to emotion perception deficits [62,63]. In other words, eating disorder behaviours would offer short-term relief to the detriment of more adaptive strategies.

We identified only one study on the association between TEI and AN in youth, and this was because current data mostly concentrate on adult patients with AN. Since the role of emotional regulation in the onset of anorexic behaviours is grounded early in development, EI’s predictive role in the onset of AN should be more extensively examined in children and adolescents.

Remarkably, the map failed to identify any study investigating the construct of AEI concerning the onset of obesity and EDs in children and adolescents. Considering the high prevalence of obesity in children worldwide [64], and the available evidence suggesting that EDs often develop during adolescence as an attempt to control weight and to deal with a negative body image resulting from overweight problems in childhood [65], further research in the field is urgently needed.

Study investigating EI’s role (both TEI and AEI) in predicting the onset of binge eating was also absent across age groups. This absence may be because research focusing on adult patients with BED suggests that the onset of binge eating problems generally occurs later in life [66] due to repeated failed attempts to control the diet [67,68]. Still, research demonstrates that binge eating can occur both before the onset of dieting behaviour, as mostly observed in patients with AN [69,70], and after the start of dieting behaviour [71,72], as usually reported by patients with BN. Prospective risk factor studies have also found potential antecedents to binge eating in children, thus validating the role of negative affectivity in the early development of binge eating problems [51,73]. In the light of the substantial evidence that vulnerability to obesity is a risk factor for binge eating and forthcoming eating and weight disorders, EI’s role in predicting binge eating in children and adolescents is worth investigating.

The literature on developmental psychology reveals that emotional abilities constitute an important topic throughout childhood, adolescence, and adulthood. A deficit in EI might interfere with these transitions in a way that might increase the prevalence and intensity of ED symptoms across developmental stages [45].

### 4.1. Strengths and Limitations

The present systematic mapping review has some risks and limitations. One of the risks is related to selective reporting bias [30]. Four different databases providing a comprehensive list of articles were used as the source for the search process to lessen this problem. Moreover, the criteria used to select the articles to be examined during the study (selection bias) could have further affected the present work results. The decision to limit the search only to cross-sectional or cohort studies may have led to the exclusion of some significant contributions. It is also worth noting that it was decided to exclude grey literature (e.g., theses, internal reports) from the study. This may have affected the validity of the study, but grey literature is not usually subjected to a rigorous review process. Both the inclusion and exclusion criteria were clearly defined to reduce such risks.

Moreover, the risk of inaccuracy in the data extraction and misclassification was mitigated as these processes were performed independently by two authors. A consensus decision, including a third author, was made in the case of no agreement. Lastly, although the mean age of the participants included in the study by Markey et al. (2007) [35] was slightly above the threshold, it was retained due to its significant finding and keeping in mind the World Health Organization’s definition for adolescents as ‘individuals in the 10–19 years age group’ [74].

### 4.2. Future Research and Practical Directions

Research widely demonstrates that EDs and overweight problems have high co-occurrence levels and cause significant distress throughout the lifespan [75]. It is, therefore, important to better understand the shared risk factors for both outcomes. In this respect, individuals with obesity and EDs typically report difficulties in emotional recognition and regulation [76], which may be united by a growing literature exploring the EI construct.

Research has proved that the best possibility to build lifelong needed emotional-social skills is in early childhood. Research also suggests that the ability to deal with one’s own and others’ emotions developed at this stage, in turn, leads to the formation of a rigid modality to interact with oneself, others and the world, which affects individuals’ personalities and lifestyles, including eating patterns [77].

In this scenario, a supportive relationship with parents can help to buffer stressful experiences. Parents are often essential role models for eating behaviours and physical exercise aptitudes in children [1,3], and good parenting skills become particularly crucial during an early stage in life.

To date, although studies broadly demonstrate the existence of an association between EI (trait and ability) and obesity/EDs [78], findings on the predictive role of EI in the onset of eating and weight disorders in childhood and young adults are scant and questionable.

Further investigations aimed explicitly at understanding the prognostic role of EI, alone and/or in relation with other established predictors of eating and weight disorders (i.e., body image disturbances), in childhood are required for a better understanding of the mechanism involved in the aetiopathogenesis of eating problems and the development of effective preventive actions of care.

Additional investigations comparing clinical and non-clinical samples of participants in different developmental stages are needed to this end. There is a body of research examining the role of eating behaviours in non-clinical samples to detect essential differences in the dynamics of personal factors associated with ED behaviour. However, there is a lack of studies comparing non-clinical samples with clinical samples.

Moreover, to properly explore connections between variables over time and to assume cause-and-effect relationships between variables, longitudinal study designs are strongly required to test EI’s impact on eating attitudes in both non-clinical and clinical contexts.

Findings from this study also highlight the need for a more comprehensive conceptual validation of EI in the study of ED risk factors and symptoms and the development of more straightforward and more effective EI measurement instruments in young people [79]. Accurate EI measures would provide clinicians with a potential intervention guide for screening, prevention and treatment programmes based on EI to minimise the risk of ED in populations potentially at risk, such as children with obesity.

## 5. Conclusions

This mapping review aimed to take the first steps in studying the relationship between EI and EDs. Significant correlations highlighted between the two constructs confirm the hypothesis that high TEI levels could act as a protective factor in the development of eating and weight problems and body image dissatisfaction. However, inconsistencies across studies create challenges in interpreting and generalising these indications across different stages of development. Future longitudinal studies might be useful for a theoretical understanding of EI’s nature and its associations with various eating-related behaviours to guide prevention programmes for young people at risk for eating and weight disorders.

## Figures and Tables

**Figure 1 ijerph-18-02054-f001:**
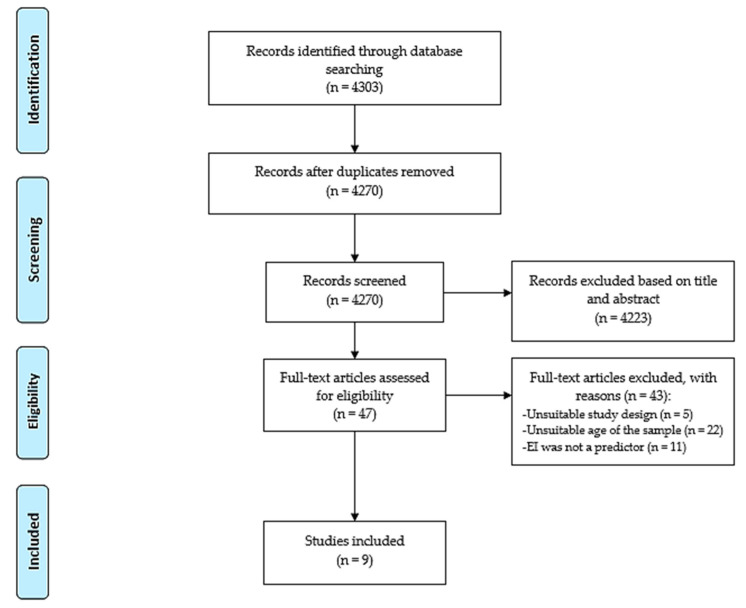
PRISMA flow chart.

**Figure 2 ijerph-18-02054-f002:**
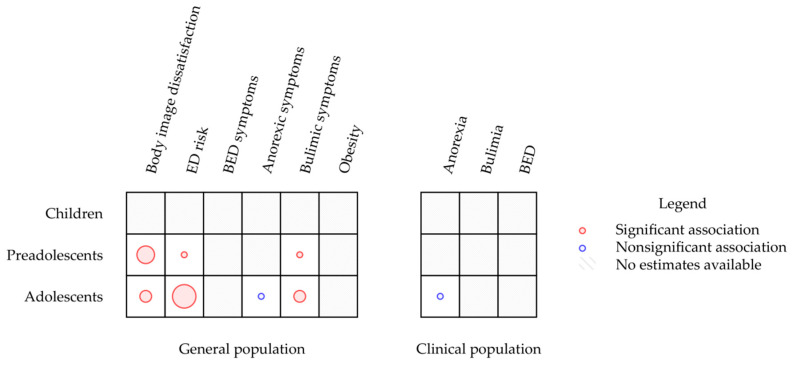
Mapping diagram of examined studies. Note. The map synthesises estimates of the association between EI and eating- and weight-problem-related constructs. Different estimates calculated by single studies are reported separately. The radius of circles is proportional to the number of estimates available regarding the association between EI and the relative construct.

**Table 1 ijerph-18-02054-t001:** Description of the included studies (*n* = 9).

Author, Year	Country	EI Measure	Study Aim	Primary Outcome	Outcome Measure	Population	Sample Size	Female %	Mean Age (SD)	Age Range	Mean BMI (SD)
Amado Alonso, 2020 [34]	ES	EQI-YV	To explore how body image satisfaction and gender act as modulators of EI	Body image satisfaction	Stunkard Figure Rating Scale	Preadolescents (non-clinical)	944	42	10.76 (1.11)	9–12	N/A
Cuesta-Zamora, 2018 [42]	ES	TEIQue-ASF	To explore the relationship between EI and ED symptoms	Body dissatisfaction, bulimic symptoms, drive for thinness	EDI-3 subscales: DT-EDI-3, B-EDI-3, BD-EDI-3	Preadolescents (sample 1) and adolescents (sample 2) (non-clinical)	762	Sample 1: 51.8; Sample 2: 47	Sample 1: 10.55 (0.60); Sample 2: 13.53 (1.25)	Sample 1: 10–12; Sample 2: 12–17	Female: 19.2 (3.9); Male: 19.1 (3.6)
Li, 2018 [39]	CN	WLEIS	To explore the relationship between EI, social anxiety and ED risk	ED risk	EAT-26	Adolescents (non-clinical)	784	51.3	17.12 (1.32)	15–20	21.47 (2.39)
Li, 2019 [40]	CN	WLEIS	To explore the role of EI in moderating (1) the relationship between body esteem and ED and (2) the mediating effect of social appearance anxiety between body esteem and ED	ED risk	EAT-26	Adolescents (non-clinical)	2509	54.1	16.12 (1.45)	15–20	N/A
Markey, 2007 [35]	US	EQI-YV-SF	To explore the role of EI, alexithymia and coping strategies in moderating the relationship between a negative affect and ED symptoms	Bulimic symptoms	Bulimia test, revised	Adolescents (non-clinical)	154	100	18.66	17–23	23.2
Peres, 2017 [36]	FR	EQI-YV	To explore the differences in EI, empathy and alexithymia between adolescents with AN and healthy controls	AN	DSM-IV Mini-International Neuropsychiatric Interview	Adolescents (clinical vs. non-clinical)	79	100	AN: 16.2 (1.44); HC: 16.4 (1.73)	AN: 13.1–18.9; HC: 13.1–18.8	N/A
Pollatos, 2020 [38]	DE	EQI-YV-SF	To explore the relationship between EI and body image	Body image perception	Body Silhouette Chart	Preadolescents (non-clinical)	991	49.7	9.58 (0.62)	8–11	* 17.34 (2.6)
Wong, 2014 [37]	TW	AEIS	To explore the relationship between EI and ED risk	ED risk	EAT-26	Adolescents (non-clinical)	1028	24.4	16.1 (0.7)	14–18	N/A
Zavala, 2018 [41]	MX	EQI-YV	To explore the relationship between EI and ED symptoms	ED symptoms	Millon Adolescent Clinical Inventory (MACI), eating disorders subscale	Adolescents (non-clinical)	829	52.5	13.6 (0.64)	13–15	N/A

Note. * Standardised body mass index (BMI SDS). Abbreviations: CN: China; DE: Germany; ES: Spain; FR: France; MX: Mexico; TW: Taiwan; US: United States; AN: anorexia nervosa; ED: eating disorder; EI: emotional intelligence; BMI: body mass index; SD: standard deviation; EQI-YV: Bar-On Emotional Quotient Inventory: Youth Version; EQI-YV-SF: Bar-On Emotional Quotient Inventory: Youth Version—Short Form; TEIQue-ASF: Trait Emotional Intelligence Questionnaire—Adolescent Short Form; WLEIS: Wong and Law Emotional Intelligence Scale; AEIS: Adolescent Emotional Intelligence Scale.

**Table 2 ijerph-18-02054-t002:** Methodological quality and summary of results of the included studies (*n* = 9).

Author, Year	Study Design	Selection	Measurement	Reporting	Confounding	Results
Amado Alonso, 2020 [34]	+	+	+	+	+	Body image satisfaction significantly correlates with interpersonal, stress management, adaptability, and mood components, but not with the intrapersonal component, of EI. The association between the stress management component of EI and body image satisfaction is significant only for boys.
Cuesta-Zamora, 2018 [42]	+	−	+	+	+	EI significantly and negatively correlates with ED scores in both samples.
Li, 2018 [39]	−	+	+	+	+	EI negatively correlates with social anxiety, which results in partially mediating the relationship between EI and ED risk.
Li, 2019 [40]	−	+	+	+	+	Social appearance anxiety partially mediates the relationship between body esteem and ED risk, whilst EI moderates the effects of body esteem on social appearance anxiety and ED.
Markey, 2007 [35]	−	−	+	+	+	EI significantly predicts bulimic symptoms. EI, alexithymia and coping strategies do not moderate the relationship between a negative affect and bulimic symptoms.
Peres, 2017 [36]	+	−	+	+	+	Samples (clinical vs. non-clinical) significantly differ for intrapersonal and general mood components of EI, while no differences in interpersonal, adaptability and stress management components of EI are found between groups. After controlling for anxiety and depression, no significant correlation is found between AN symptoms and the intrapersonal component of EI, whereas the relationship between the mood component of EI and AN remains significant.
Pollatos, 2020 [38]	+	+	+	+	+	Significant inverse associations are found in both the male and female subsamples between body image dissatisfaction and EI after controlling for the BMI.
Wong, 2014 [37]	+	+	+	+	−	ED symptoms positively correlate with emotional perception, emotional expression, and emotional application components, but not with the emotion regulation component, of EI.
Zavala, 2018 [41]	+	+	+	+	+	The intrapersonal, stress management and adaptability components of EI have a weak but significant inverse correlation with ED symptoms. In a multivariable model controlling for sex, only the stress management component of EI remained associated with ED risk.

Note. Plus (+) signs indicate a low risk of bias, whereas minus (−) signs indicate a high risk of bias. Only results referring to eating and weight problems are reported. Abbreviations: EI: emotional intelligence; ED: eating disorder; AN: anorexia nervosa.

## Data Availability

Data is contained within the article or Appendix A.

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
