# Peer review of "The Relationship between Emotional Intelligence, Obesity and Eating Disorder in Children and Adolescents: A Systematic Mapping Review"

_ijerph, 2021, doi:10.3390/ijerph18042054_

Round 1

Reviewer 1 Report

This paper is aimed to provide an evidence map on the relationship 75 between EI and eating and weight disturbances in disturbances among clinical and non-76 clinical populations of children and adolescents.

From the policy implication view, we need the authors to comment on some implications for behavior change of children with ED.  For example, Amado Alonso (2020) concludes that body image satisfaction significantly correlates with interpersonal, stress management, adaptability, and mood components. Then, what kind of suggestions could be given to parents who has children with ED would be a natural question.

Author Response

Thanks for such positive comments to our paper. 

Regarding "From the policy implication view, we need the authors to comment on some implications for behavior change of children with ED.  For example, Amado Alonso (2020) concludes that body image satisfaction significantly correlates with interpersonal, stress management, adaptability, and mood components. Then, what kind of suggestions could be given to parents who has children with ED would be a natural question." 

We have addressed this suggestion in the paper

Reviewer 2 Report

Dear Authors,

Thank you for submitting the manuscript: The relationship between emotional intelligence, obesity, and eating disorder in children and adolescents: a systematic mapping review. I hope you find the following commentary constructive:   Introduction line 76: Please check the sentence in terms of repeating the word "disturbances".  

General remark: After reading the introduction, I am a little confused what the Authors mean by "eating disorder". Do they mean an eating disorder like anorexia etc as the medical term used suggests? Or dietary abnormalities, such as excessive caloric/sugar/fat intake, which may be suggested by the term "eating disturbances" (line 74). In my opinion, this requires further clarification in this section of the manuscript. Especially that some of the eating disorders may be more typical for the younger age group (e.g. pica) and others for older children and adolescents (e.g. AN, BN).  

Materials and Methods:  

lines 79-85 This part in my opinion belongs more to Introduction, as an element of justification for undertaking research and selection of methodology.   

line 101: No language restrictions is rather unusual. Are publications in a language other than English, Italian, German or Spanish actually found and later eventually included?   Were obesity criteria for the population of children and adolescents up to 18 years of age the same in the analyzed studies? The use of different criteria (WHO, IOTF, ect) is a common problem when comparing studies from different centers.   

Results:

Tabele 1 The mean age of participants in Markey study is 18.66, but in Inclusion and exclusion criteria (lines 99-100) is stated that: "in a sample of participants 99 with a mean age ≤ 18 years (children; preadolescents, adolescents) were included" Could Authors refer to that?  

Discussion:

I like this part. Please add information on age for "very young children" in line 107, p8. Could the Authors refer to the lower age limit for diagnosing eating disorders (analyzed in this study) in children, based on the literature? From what age can EI (two types) be assessed?       

Author Response

Thanks for your suggestion and carefully read of the paper. We have addressed every single suggestion and amendments that you kindly provided to us.

Introduction line 76: Please check the sentence in terms of repeating the word "disturbances". 

We changed "disturbances" by "disorders", which is a more accepted term.

General remark: After reading the introduction, I am a little confused what the Authors mean by "eating disorder". Do they mean an eating disorder like anorexia etc as the medical term used suggests? Or dietary abnormalities, such as excessive caloric/sugar/fat intake, which may be suggested by the term "eating disturbances" (line 74). In my opinion, this requires further clarification in this section of the manuscript. Especially that some of the eating disorders may be more typical for the younger age group (e.g., pica) and others for older children and adolescents (e.g., AN, BN). 

AGREE WITH THE REVIEWER AND FIXED THROUGHOUT THE DOCUMENT

Materials and Methods: 

lines 79-85 This part in my opinion belongs more to Introduction, as an element of justification for undertaking research and selection of methodology.  

AGREE, WE SET IT IN THE INTRODUCTION

line 101: No language restrictions is rather unusual. Are publications in a language other than English, Italian, German or Spanish actually found and later eventually included?   Were obesity criteria for the population of children and adolescents up to 18 years of age the same in the analyzed studies? The use of different criteria (WHO, IOTF, ect) is a common problem when comparing studies from different centers.  

WE INCLUDED SPECIFICALLY THESE SUGGESTIONS IN THE PAPER

Results:

Table 1 The mean age of participants in Markey study is 18.66, but in Inclusion and exclusion criteria (lines 99-100) is stated that: "in a sample of participants 99 with a mean age ≤ 18 years (children; preadolescents, adolescents) were included" Could Authors refer to that? 

WE INCLUDED IT DUE TO THE SD WAS LOW AND PRETTY CLOSE OF OUR INCLUSION CRITERIA

Discussion:

I like this part. Please add information on age for "very young children" in line 107, p8. Could the Authors refer to the lower age limit for diagnosing eating disorders (analyzed in this study) in children, based on the literature? From what age can EI (two types) be assessed?       

THANKS FOR THE POSITIVE INPUT. WE CLARIFIED THIS POINT IN THE NEW MS

Limitations

-The discussion chapter needs to be reorganized and rewritten.

WE REORGANISED AND REWRITTEN THE LIMITATIONS SECTION

In my opinion, the study is very interesting and worth being published. In fact, the article is well-structured, the method used is congruent with the purpose of the stud and the results are presented clearly. Nevertheless, I feel I can give the following suggestions:

  1. A PRISMA flow diagram must be improved. So, please improve the quality of figure proposed that would help to improve the readability for the general journal readers. WE ARE GOING TO UPLOAD TWO ORIGINAL FIGURES WITH OPTIMAL QUALITY TO BE PRINTED AND EDITED
  2. I am afraid that the discussion chapter still needs work. I here offer some insight into how the discussion part should be re-written. REALLY APPRECIATED THE SUGGESTIONS AND WE FOLLOWED YOUR DESCRIPTIONS MENTIONED BELOW  

- First of all, I suggest including one short paragraph summarizing the purpose of the study.

- Second, I think that the whole discussion needs to be restructured to include: theoretical implications of the study, practical implications, limitations, future studies, and a brief conclusion.

I think the authors can easily follow the suggestions I have given in this review and make a new version of their interesting paper.

Reviewer 3 Report

Strengths:
- Nice description why this is a relevant topic and what the theory behind the research questions is.
- The Research Questions are clearly described.
- Good Method section.
- The description of how the data were analyzed is clarity.
- Conclusions supported by data.

Limitations
-The discussion chapter needs to be reorganized and rewritten.

In my opinion, the study is very interesting and worth being published. In fact, the article is well-structured, the method used is congruent with the purpose of the stud and the results are presented clearly. Nevertheless, I feel I can give the following suggestions:

1. A PRISMA flow diagram must be improved. So, please improve the quality of figure proposed that would help to improve the readability for the general journal readers.

2. I am afraid that the discussion chapter still needs work. I here offer some insight into how the discussion part should be re-written.

- First of all, I suggest to include one short paragraph summarizing the purpose of the study.

- Second, I think that the whole discussion needs to be re-structured to include: theoretical implications of the study, practical implications, limitations, future studies and a brief conclusion.

I think the authors can easily follow the suggestions I have given in this review and make a new version of their interesting paper.

All best wishes.

Author Response

Thanks for the positive outputs

-The discussion chapter needs to be reorganized and rewritten.

In my opinion, the study is very interesting and worth being published. In fact, the article is well-structured, the method used is congruent with the purpose of the stud and the results are presented clearly. Nevertheless, I feel I can give the following suggestions:

  1. A PRISMA flow diagram must be improved. So, please improve the quality of figure proposed that would help to improve the readability for the general journal readers. WE UPLOADED TWO NEW FILES WITH A BETTER QUALITY TO BE EDITED AND PRINTED

2. I am afraid that the discussion chapter still needs work. I here offer some insight into how the discussion part should be re-written.

- First of all, I suggest to include one short paragraph summarizing the purpose of the study.

  • Second, I think that the whole discussion needs to be re-structured to include: theoretical implications of the study, practical implications, limitations, future studies and a brief conclusion.

WE FOLLOWED YOUR RECOMMENDATION AND IMPROVED THESE SECTIONS AND WRITTINGS